# Lagging behind in health and work expectancies: Increasing disadvantage of individuals with lower educational level in Germany

Juliane Tetzlaff[1]*, Fabian Tetzlaff[2], Marc Luy[3]

1 Medical Sociology Department, Hannover Medical School, Hanover, Germany, 2 Division of Social Determinants of Health, Department of Epidemiology and Health Monitoring, Robert Koch Institute, Berlin, Germany, 3 Vienna Institute of Demography of the Austrian Academy of Sciences, Vienna, Austria

* Tetzlaff.Juliane@mh-hannover.de

## Background

Many governments increased the retirement age in response to population ageing. Against this backdrop, it remains unclear whether the development in healthy life years can keep pace with the increase in working life years and whether people with lower socio-economic status are left behind. We investigated time trends in healthy life years and healthy working life years and how trends differ between educational groups in Germany.

## Methods

Temporal trends in partial life expectancy between age 30 and 69 were assessed using data from the German Socio-Economic Panel (GSOEP, N=40,150) of three educational groups. Based on this, education-specific (Un)Healthy Life Expectancy ((U)HLE) and (Un)Healthy Working Life Expectancy ((U)HWLE) were calculated using the Sullivan method. Health is assessed on the basis of two health indicators: the physical and the mental score of health-related quality of life (p/mHRQoL). Both has been shown to be important indicators for working-age health.

## Results

With respect to pHRQoL, HLE increased among men and women with higher educational attainment while it decreased in men with lower educational level. HWLE increased stronger in men and women with higher than with lower educational attainment. UHWLE increased strongest in persons with lower educational attainment. In terms of mHRQoL, HLE increased in all educational groups except for the group of women with lower educational attainment. UHLE decreased among men and women with middle and higher educational level. HWLE increased in all groups, with increases being strongest among higher educated individuals. UHLE increased in women with lower educational attainment but decreased in men and women with higher educational level.

**Data availability statement:** For this paper the data of the German Socio-Economic Panel Study carried out at the DIW (Deutsches Institut für Wirtschaftsforschung) (GSOEP V.38) was used. The authors are not permitted to share the data underlying this study. The GSOEP data belong to DIW, are only available upon request to the data owner (soepmail@diw. de), and a contract must be concluded between the DIW and the researcher before access to the data is granted.

**Funding:** The author(s) received no specific funding for this work.

**Competing interests:** The authors have declared that no competing interests exist.

## Discussion

We found polarising trends, with healthy life years and healthy working years developing less favourably among people with lower than with higher educational level. This applies to both the physical and mental component of HRQoL. The study shows that people with lower educational level are less able to keep pace with the prolonged working life from a health perspective and that more effective prevention is needed to stop the widening of health inequalities in working age.

## Introduction

In many high-income countries, governments have been reacting to their ageing populations and the growing imbalance between workers and non-workers by increasing the statutory retirement age. This led to prolonged working lives. With a current average age of 44.7 years [1], Germany is one of the oldest populations in the world. A closer look at Germany therefore promises interesting insights, as it already has a comparatively high retirement age (currently age 66), which has already been decided to be raised to the age 67 in the coming years [2]. However, there is still little research on whether working lives should be extended from a health perspective [3,4]. It is still largely unclear whether the changes in healthy life years can keep pace with the increase in working lifetime and whether people with a low socio-economic status (SES) are left behind. The aim of our paper is to narrow this research gap. We analyse social inequalities in health and work expectancies in the German working-age population and its development over time.

Poor health hampers the ability to work and it leads to early labour market exits [5]. Therefore, the question of how healthy life years and healthy working life years developed over time is crucial considering the current debate on (further) increases in retirement age. Two indicators are helpful to answer this question: The first is Healthy Life Expectancy (HLE) which depicts the number of years spent in good health irrespective of the labour force status [6]. Accordingly, HLE is suitable to investigate how health expectancies have developed in the working-age population. Previous research has therefore considered HLE as potential to increase the length of working lives [7]. The second is Healthy Working Life Expectancy (HWLE). This indicator is a specification of HLE. It was introduced to investigate the number of years spent working and in good health [8], and is thus focused on the health of the working population. Whereas HLE separates the total number of life years into those spent in good and poor health, HWLE is restricted to the working life years.

Recent studies focusing on health among the German working-age population provided a mixed picture and reported either increasing, constant, or even decreasing rates of good health depending on the age group and health indicator considered [9–14]. Research on social inequalities in HWLE is still very limited [4,15]. Using different health indicators, most studies found increasing HWLE over time [11,12,16–20]. However, this increase was often accompanied by an increase in working years in poor health [11,12,17]. Furthermore, international studies reported

substantial inequalities by SES in HWLE, pointing towards less healthy working years among persons with lower SES [16,21–23]. Only very few of them also investigated trends in social inequalities in the older working-age population and found increasing inequalities over time [16,20,24].

The reason for the lack of studies on social inequalities in Health and Work Expectancies in the German population is that official statistics do not provide data on mortality by SES. To the best of our knowledge, there are only two studies on time trends in HLE in Germany, which investigated educational inequalities. These studies used mortality information from survey data. The first study focused on trends in long-term care-free LE (CFLE) among men at age 65 between 1997 and 2012. This study found widening educational inequalities in CFLE [25]. The second study reported HLE at age 40 and found narrowing educational inequalities in HLE between 1989 and 2009 in women and rather constant inequalities among men [26]. However, there is still a lack of studies that focus specifically on working age.

Due to the lack of mortality data by SES in the official statistics, previous studies on time trends in HWLE and WLE had to ignore mortality and therefore to exclude the population aged 65 years and older. These studies provided valuable insights into the development of working life expectancies in the German population over time [20,27]. However, against the backdrop of the ongoing debate about a further increase in the statutory retirement age beyond the age of 67, the inclusion of people over the age of 64 promises a better understanding of the possibilities and limits of a further extension of working life. For such an analysis, the inclusion of mortality appears reasonable. If mortality is ignored it can be assumed that social inequalities in H(W)LE are somewhat underestimated, since social inequalities in the underlying life expectancy (LE) are not taken into account. This issue is likely to be exacerbated if the age range included in the analysis is extended to higher age groups with higher mortality rates (e.g., above age 64), which are the target of the ongoing discussion on retirement age. Together with the substantial inequalities in mortality in the German population [28–30], it appears reasonable to consider developments in health and work expectancies including mortality whenever possible. Ignoring inequalities in mortality is therefore sub-optimal, but often the only possibility to investigate inequalities in HWLE in the German population. Due to these data limitations, there is so far only one study on HWLE in Germany ignoring mortality, which reported increasing educational inequalities in HWLE over time [20].

### Our paper addresses the following questions

-Are there educational inequalities in HLE and HWLE among the working-age population? How large are these inequalities?
-How did educational inequalities in HLE and HWLE in the working-age population develop over time?

## Methods

### Data

The study is based on the data of the German Socio-Economic Panel (GSOEP V.38) of the waves 2000–2020. The GSOEP is one of the largest surveys on health, employment, and other SES characteristics of persons living in private households with currently about 30,000 interviewed persons aged 18 years and older per year in Europe [31]. If a household is selected, all household members aged 18 years and older are asked to participate [31]. The survey was carried out by the German Institute for Economic Research (Deutsches Institut für Wirtschaftsforschung, DIW). The data are available for scientific purposes on request from the DIW. The DIW is responsible for all legal requirements and ethical aspects regarding these data. Therefore, it was not necessary for the authors to obtain the approval of an ethics committee or informed consent of the participants for this study. All data were fully anonymized when we received them.

The dataset contains information on the latest whereabouts of former participants (deceased, living in Germany, living aboard, unknown). This information were either drawn from household members still participating in the survey, obtained during the field work, or from one of rigorous drop-off studies performed by the DIW in 2001, 2006, and 2008. During

these drop-out studies, the residents' registration offices were contacted to determine the current vital status of participants who have left the panel or who have not responded to the interview request. Based on drop-out studies, the vital status of about 90% of the former participants could be identified. As the last drop-out study was conducted in 2008, the underreporting of deaths can assumed to be higher in later years [32].

## Statistical analysis

### Partial life expectancy

Using this information on current and former participants' survival status, partial life expectancy (PLE) for the ages 30–69 for the total population and education subgroups was calculated as a first step. PLE represents the expected years of life in a certain age range, i.e., PLE between ages 30 and 69 could be a maximum of 40 years if mortality between the ages of 30 and 69 was zero. Previous studies have used estimation-based life table calculations to determine SES inequalities in LE based on the GSOEP assuming constant relative social inequalities across age [33]. While this method has the advantage of working well with lower case numbers, we prefer to apply the "classic" life table calculation based on observed mortality rates, allowing for flexible SES disparities in mortality across age. Following this approach, sufficient case numbers are needed. Therefore, we combined the data of nine survey waves (2000–2008, 2008–2016, 2012–2020) to calculate PLE for the middle year of the respective period of nine calendar years based on the mortality rates of five-year age groups. Inequalities in mortality by education were assessed for the study population at age 30 up to age 69 years to cover a broad range of (future) working age. Within this age range, educational level can be assumed to remain constant and the numbers of deaths within educational subgroups were sufficient to apply classic period life tables calculations.

Educational inequalities in PLE were assessed using three groups that reflect the number of years of schooling usually needed to obtain the respective school-leaving certificate. The lower educational group includes all persons with no more than 9 years of schooling (max. Hauptschule), which also covers persons without a school-leaving certificate. The middle group comprises those with 10–11 years of schooling, which corresponds a comprehensive school-leaving certificate (Realschulabschluss). The group of higher-educated individuals represents those with 12–13 years of schooling, which were qualified for studying at a university ((Fach-)Abitur).

The GSOEP data is affected by the well-known health-related non-response in surveys and underreporting of deaths [32]. Consequently, the total and education-specific LE of the survey participants is slightly higher than that of the whole German population [32]. To derive estimates that are in line with the official values for LE in Germany, we adjusted the PLE estimated from GSOEP values by relating them to the corresponding PLE values of the total German population using the adjustment factor $Y$ calculated by:

$$Y = \frac{PLE_{GER}}{(PLE_{GSOEP,E1}*w1 + PLE_{GSOEP,E2}*w2 + PLE_{GSOEP,E3}*w3)},$$

with $E_i$ representing the three educational groups and $p_i$ being the proportion of the educational group in the total GSOEP survey (corresponding case numbers can be found in Table 1). Thus, we composed the total PLE from the specific PLEs of its educational subgroups. Following this formula, Y depicts the relative difference between PLE of the total GSOEP population and the total German population. $Y$ was then used to adjust each of the education-specific PLEs. By doing so, we assumed that the underestimation of mortality in the GSOEP does not differ between the three educational groups. According to these calculations, the PLE for ages 30–69 of the total GSOEP sample is on average 0.5 years higher in men and 0.2 years higher in women than those of the total German population.

### Health and work expectancy

Based on the PLE, Healthy Life Expectancy (HLE) and Healthy Working Life Expectancy (HWLE) were calculated with the Sullivan method [34]. This method combines the total number of life years lived derived from period life table and

**Table 1.** Characteristics of the study population.

| | Number of Individuals | Healthy pHRQoL | Healthy working pHRQoL | Healthy mHRQoL | Healthy working mHRQoL | Number of Deaths | Partial Life Expectancy between ages 30–69 | |
|---|---|---|---|---|---|---|---|---|
| | N | (in %) | (in %) | (in %) | (in %) | | | (95%-CI) |
| **Men** | | | | | | | | |
| **Total** | | | | | | | | |
| 2004 | 10,131 | 84.6 | 66.3 | 85.0 | 62.9 | 501 | 37.4 | (37.2-37.6) |
| 2012 | 14,793 | 83.7 | 69.7 | 85.5 | 67.7 | 368 | 37.7 | (37.5-38.0) |
| 2018 | 15,226 | 84.0 | 72.0 | 87.1 | 72.0 | 344 | 37.9 | (37.7-38.1) |
| **Lower Educational Attainment** | | | | | | | | |
| 2004 | 4,001 | 78.0 | 54.3 | 84.9 | 54.4 | 335 | 36.7 | (36.3-37.2) |
| 2012 | 5,285 | 76.6 | 57.3 | 83.6 | 58.4 | 199 | 37.2 | (36.9-37.6) |
| 2018 | 5,315 | 73.9 | 57.8 | 84.9 | 62.6 | 178 | 37.4 | (37.1-37.8) |
| **Middle Educational Attainment** | | | | | | | | |
| 2004 | 2,348 | 88.3 | 75.4 | 85.5 | 70.5 | 64 | 37.6 | (37.1-38.0) |
| 2012 | 3,588 | 84.8 | 74.2 | 85.5 | 71.6 | 79 | 37.5 | (37.0-38.0) |
| 2018 | 2,755 | 84.4 | 73.9 | 88.0 | 74.5 | 73 | 37.7 | (37.3-38.2) |
| **Higher Educational Attainment** | | | | | | | | |
| 2004 | 3,756 | 91.1 | 76.4 | 84.6 | 69.1 | 93 | 38.3 | (38.0-38.6) |
| 2012 | 5,797 | 90.3 | 79.4 | 87.5 | 74.7 | 89 | 38.4 | (38.2-38.7) |
| 2018 | 6,578 | 91.9 | 82.1 | 88.4 | 78.1 | 91 | 38.4 | (38.2-38.7) |
| **Women** | | | | | | | | |
| **Total** | | | | | | | | |
| 2004 | 10,584 | 81.5 | 50.8 | 79.3 | 46.3 | 283 | 38.6 | (38.5-38.8) |
| 2012 | 17,262 | 80.8 | 58.0 | 80.6 | 54.7 | 210 | 38.8 | (38.6-38.9) |
| 2018 | 15,721 | 80.7 | 62.2 | 82.1 | 60.5 | 206 | 38.8 | (38.7-38.9) |
| **Lower Educational Attainment** | | | | | | | | |
| 2004 | 4,072 | 74.4 | 38.0 | 79.8 | 37.3 | 190 | 38.4 | (38.5-38.8) |
| 2012 | 5,248 | 72.5 | 42.9 | 79.6 | 44.1 | 112 | 38.3 | (38.6-38.9) |
| 2018 | 4,669 | 71.0 | 46.5 | 79.1 | 48.0 | 103 | 38.3 | (38.7-38.9) |
| **Middle Educational Attainment** | | | | | | | | |
| 2004 | 3,206 | 86.5 | 58.8 | 78.9 | 51.9 | 54 | 38.7 | (38.4-39.0) |
| 2012 | 5,458 | 82.6 | 62.6 | 80.5 | 58.2 | 58 | 38.8 | (38.5-39.0) |
| 2018 | 3,987 | 80.7 | 64.3 | 81.7 | 62.3 | 65 | 38.8 | (38.5-39.0) |
| **Higher Educational Attainment** | | | | | | | | |
| 2004 | 3,273 | 87.8 | 62.9 | 79.2 | 54.8 | 37 | 39.1 | (38.8-39.4) |
| 2012 | 6,432 | 88.3 | 69.8 | 81.8 | 62.7 | 38 | 39.2 | (39.0-39.4) |
| 2018 | 6,684 | 87.5 | 71.5 | 84.3 | 67.8 | 35 | 39.3 | (39.1-39.4) |

Note: Health and Work Expectancies are given as partial life expectancies at age 30 up to age 69. 95% confidence intervals are given in brackets. Data source: GSOEP 2000–2020, authors' own calculations.

the age-specific proportions of persons being in good health or being in active work and in good health to calculate the expected number of life years spent healthy or healthy working. In line with the definition of the International Labour Organization [35], a person is defined as working if he or she had any paid job during the last seven days.

In this study, H(W)LE is defined as the number of (working) years spent free of poor health based on two health indicators: physical Health-related Quality of Life (pHRQoL) and mental Health-related Quality of Life (mHRQoL). HRQoL describes the individual assessment of general health related to mental and physical functioning [36]. Analysing the two indicators makes it possible to depict physical as well as metal components of HRQoL, which are both known to be associated with poor labour market outcomes (such as low labour force participation and early retirement), and to have a substantial impact on work ability, work motivation, and sick leave [37,38]. pHRQoL and mHRQoL represent therefore important health indicators to investigate the development of the length of working life from a health perspective.

pHRQoL and mHRQoL are based on the SF-12v2 questionnaire [36,39], which is used in the GSOEP every two years since 2002. The SF-12 contains 12 health items depicting eight dimensions of HRQoL, for which a subgroup can be broken down to the norm-based physical component score (PCS) and the norm-based mental component score (MCS). The values rage from 0 to 100 and are standardised to a national norm, where 50 represents the average score in the GSOEP population in 2004 and a standard deviation of 10 marks a clear difference to the national mean [39]. Poor pHRQoL and poor mHRQoL were therefore defined for values of 40 and below, indicating a clear downward deviation from the national average score, while score values above 40 are classified as "healthy" in terms of p/mHRQoL. The age-specific proportions of healthy men or women were then combined with the life tables to calculate the PLE between ages 30 and 69 spent with good p/mHRQoL (HLE respective HWLE). The remaining life years, i.e., the difference between total PLE and HLE/HWLE, was defined as unhealthy life expectancy (UHLE) respective unhealthy working life expectancy (UHWLE). It must be kept in mind that all Health and Work Expectancies reported in this paper are partial expectancies of years spent between age 30 and 69. For the sake of readability we further refer to them simply as (U)HLEs respective (U)HWLEs either in terms of pHRQoL or in in terms of mHRQoL for the remainder of the paper.

Health and work expectancies for pHRQoL and mHRQoL were calculated for the entire study population as well as for the three educational groups based on the proportions of five-year periods (2002–2006, 2008–2014, 2016–2020) in order to determine the expectancies for the middle year of the respective period. All analyses were stratified for gender. Overall, time trends in Health and Work Expectancies between 2004 and 2018 are smooth. Therefore, the results in the figures are given for the years 2004 and 2018 only to facilitate readability. Results including the middle year 2012 can be found in the appendix (S1-S4 Tables in S1 File). Individuals with missing information on education, pHRQoL, mHRQoL, or labour status had were excluded from the analyses (14%). All analyses were carried out using Stata 14 MP and R4.1.0. The 95% confidence intervals (CI) are based on robust standard errors taking into account the variance of health and work proportions and mortality.

## Results

Across periods, about 1,900 deaths between ages 30 and 69 were recorded with lower numbers in the higher than in the lower educated group. Overall, we found clear differences in the proportion of average or good pHRQoL and mHRQoL between educational groups. PLE between ages 30 and 69 showed the expected gradient with lower values in lower educated than higher educated individuals (Table 1).

### Educational inequalities in Health and Work Expectancies

Health and Work Expectancies are largely influenced by educational attainment. This holds for both genders, both health indicators, and each year investigated (Fig 1 and 2). In 2018, HLE in terms of pHRQoL between ages 30 and 69 for men with lower educational attainment was 28.5 years and thus 6.5 years lower than that of men belonging to the higher educated group. In the same year, HLE was with 28.2 years 5.7 years lower in women with lower than with higher educational attainment. Similar inequalities emerged in UHLE between ages 30 and 69: Men and women with lower education can expect more years in poor pHRQoL in 2018 (men: 8.9, women: 10.1) than individuals with higher education (men: 3.5, women 5.4) (Fig 1). Inequalities in HLE in terms of mHRQoL are smaller but also evident. In 2018, HLE in men with

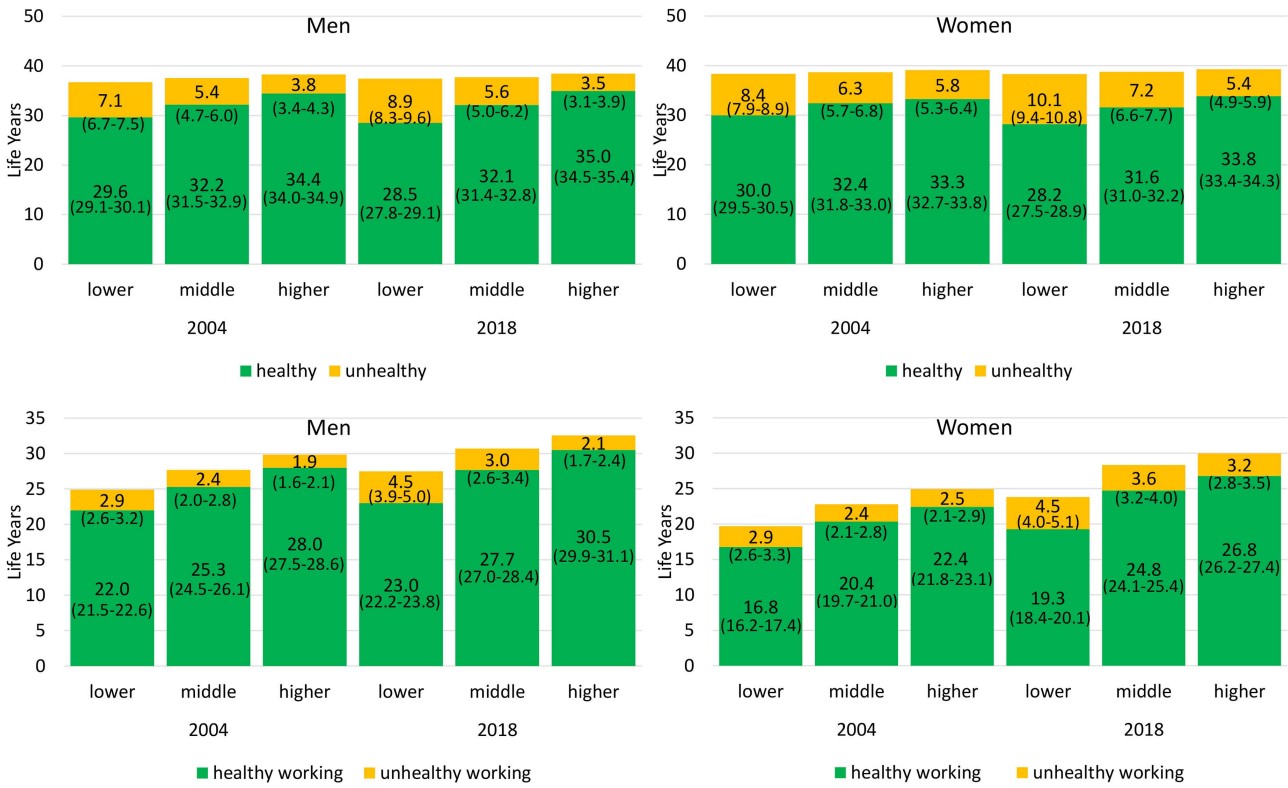

**Fig 1. Educational inequalities in partial Healthy Life Expectancy, Unhealthy Life Expectancy, Healthy Working Life Expectancy, and Unhealthy Working Life Expectancy between ages 30 and 69 in terms of Physical Health-related Quality of Life by year and gender.** Note: Health and Work Expectancies are given as partial life expectancies at age 30 up to age 69. 95% confidence intervals are given in brackets. Data: GSOEP 2000–2020, authors'own calculations.

lower education was 2.3 years higher than that of men with lower educational attainment (34.1 vs. 31.8). For women, the difference was 3.1 years (33.1 vs 30.0). UHLE in terms of mHRQoL increased over time reaching 1.4 years in men and 2.2 years in women (Fig 2).

Educational inequalities in HWLE and UHWLE were even greater: The expected years free of poor pHRQoL (HWLE) between ages 30 and 69 in 2018 in both men and women with lower educational attainment was 7.5 years lower (men: 23.0 years, women: 19.3 years) than in their counterparts with higher educational attainment (Fig 1). UHWLE in terms of pHRQol was 4.5 for both men and women with lower educational level, and 2.1 and 3.2 for men and women with a higher educational attainment. Individuals with middle educational attainment lie in between for both women and men and in all periods (Fig 1). With respect to mHRQoL, inequalities in HWLE in 2018 amounted to 4.7 years in men (29.2 vs. 24.5) and 6.1 years in women (25.6 vs. 19.5). In 2018, working life years in poor mHRQoL was about 3 years in men and 4 years in women, with no significant differences between educational groups (Fig 2).

## Time trend in educational inequalities in Health and Work Expectancies

Overall, Health Expectancies in terms of pHRQoL between ages 30 and 69 did not change much over time in the total population (Fig S1 in in S1 File), while HLE in terms of mHRQoL increased and UHLE decreased slightly (Fig S2 in S1 File). Regardless of the health indicator considered, there were considerable disparities in time trends between the educational groups. Fig 3 and 4 show in which educational groups Health and Work Expectancies have increased or decreased

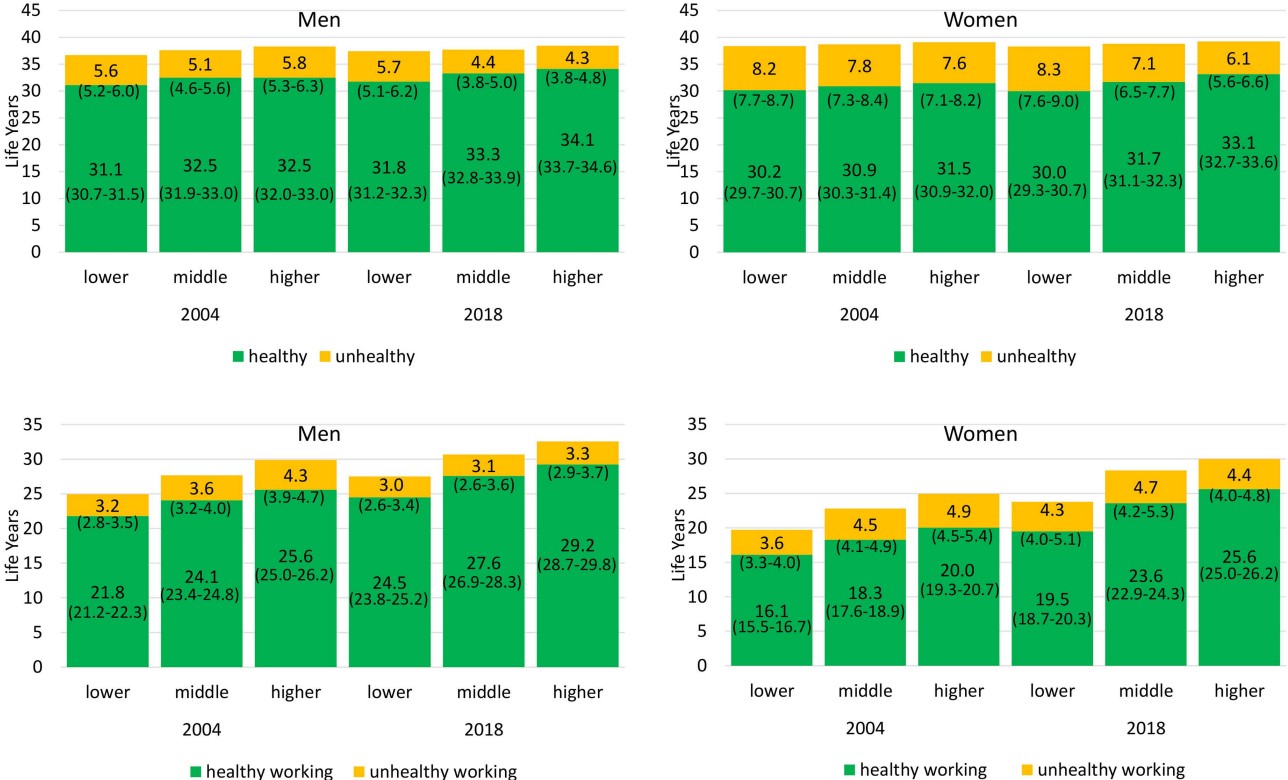

**Fig 2. Educational inequalities in partial Healthy Life Expectancy, Unhealthy Life Expectancy, Healthy Working Life Expectancy, and Unhealthy Working Life Expectancy between ages 30 and 69 in terms of Mental Health-related Quality of Life by year and gender.** Note: Health and Work Expectancies are given as partial life expectancies at age 30 up to age 69. 95% confidence intervals are given in brackets. Data: GSOEP 2000–2020, authors'own calculations.

between 2004 and 2018. Overall, inequalities widened over time: While HLE in terms of pHRQoL did not change significantly in people with higher or middle educational attainment, it decreased in men (−1.1 years) and women (−1.8 years) with lower educational attainment. During the same period, UHLE did not change significantly in men and women with higher or middle educational attainment. At the same time, unhealthy years increased in men (+ 1.8 years) and women (+ 1.7 years) with lower educational level (Fig 3). The time trends are similar for mHRQoL (Fig 4). Here too, changes over time in HLE and UHLE have widened educational inequalities. HLE increased more strongly among men and women with higher educational attainment, while UHLE declined only among men and women with higher and middle educational attainment (Fig 4).

With the exception of men with lower educational attainment, HWLE in terms of pHRQoL increased in all educational groups, albeit at different paces. This led to widening inequalities over time. The strongest increases in HWLE were found for the middle and higher educated groups: In men, HWLE increased by more than 2.4 years in these groups. In women, the pattern was similar, with increases in HWLE being overall stronger than in men: 4.4 years increases were found in women with middle and higher and 2.5 years in women with lower educational attainments. In contrast, the increase in UHWLE in terms of pHRQoL of 1.6 years for men and women was larger in the group with lower than in those with higher educational attainments (no sign. change in men and women, see Fig 3). HWLE in terms of mHRQoL increased in all educational group, with the smallest increase in men with lower educational attainment (+2.7 years vs. 3.4 to 5.6 in the other

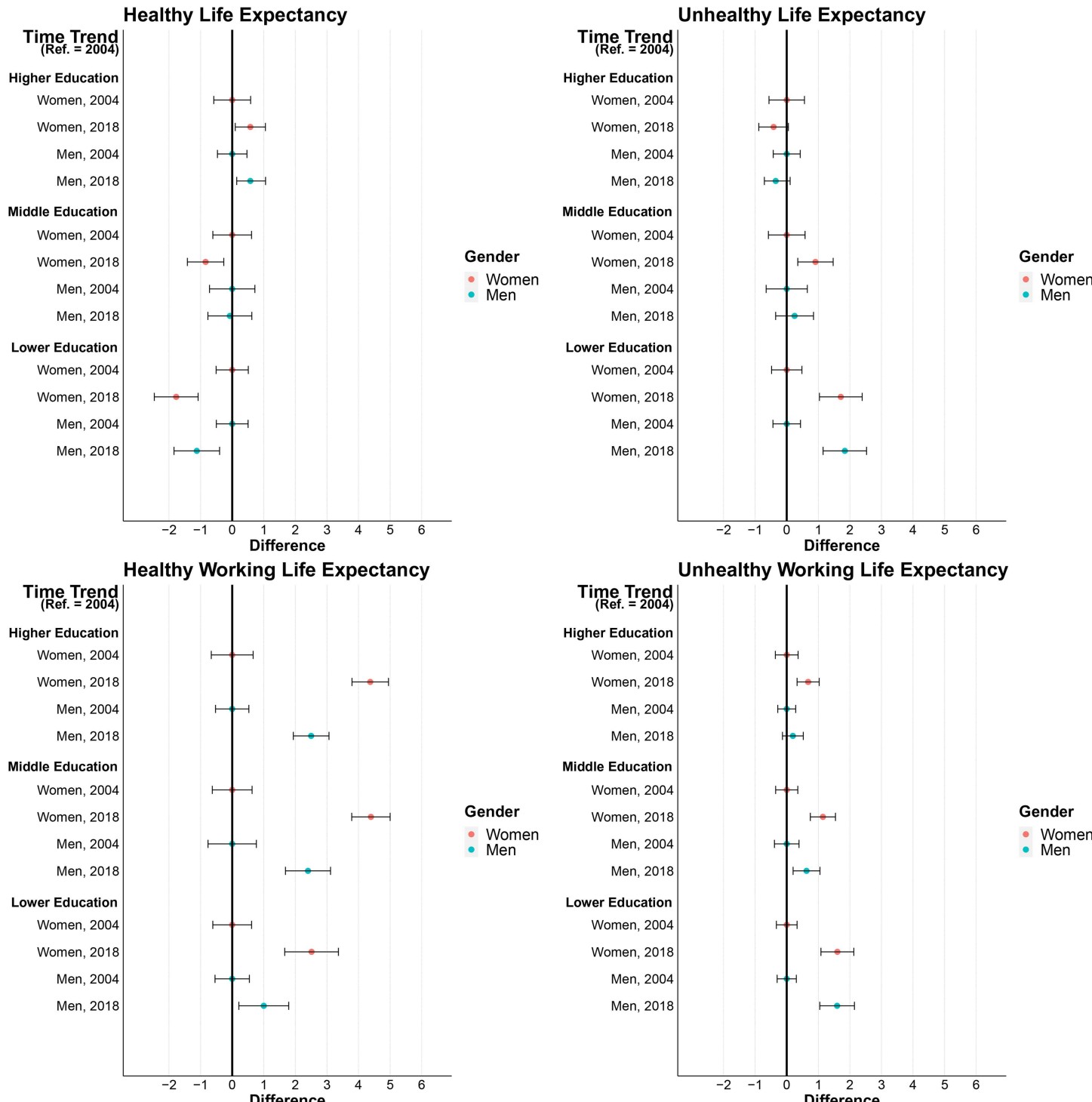

**Fig 3. Time trend in educational inequalities in partial Healthy Life Expectancy, Unhealthy Life Expectancy, Healthy Working Life Expectancy, and Unhealthy Working Life Expectancy between ages 30 and 69 in terms of Physical Health-related Quality of Life by year and gender.** Note: Health and Work Expectancies are given as partial life expectancies at age 30 up to age 69. 95% confidence intervals are given in brackets. 95% CIs were derived from the original (U)HLE and (U)HWLE but were shifted on the x-axis towards zero according to the difference value. Data Source GSOEP 2000–2020, authors' own calculations.

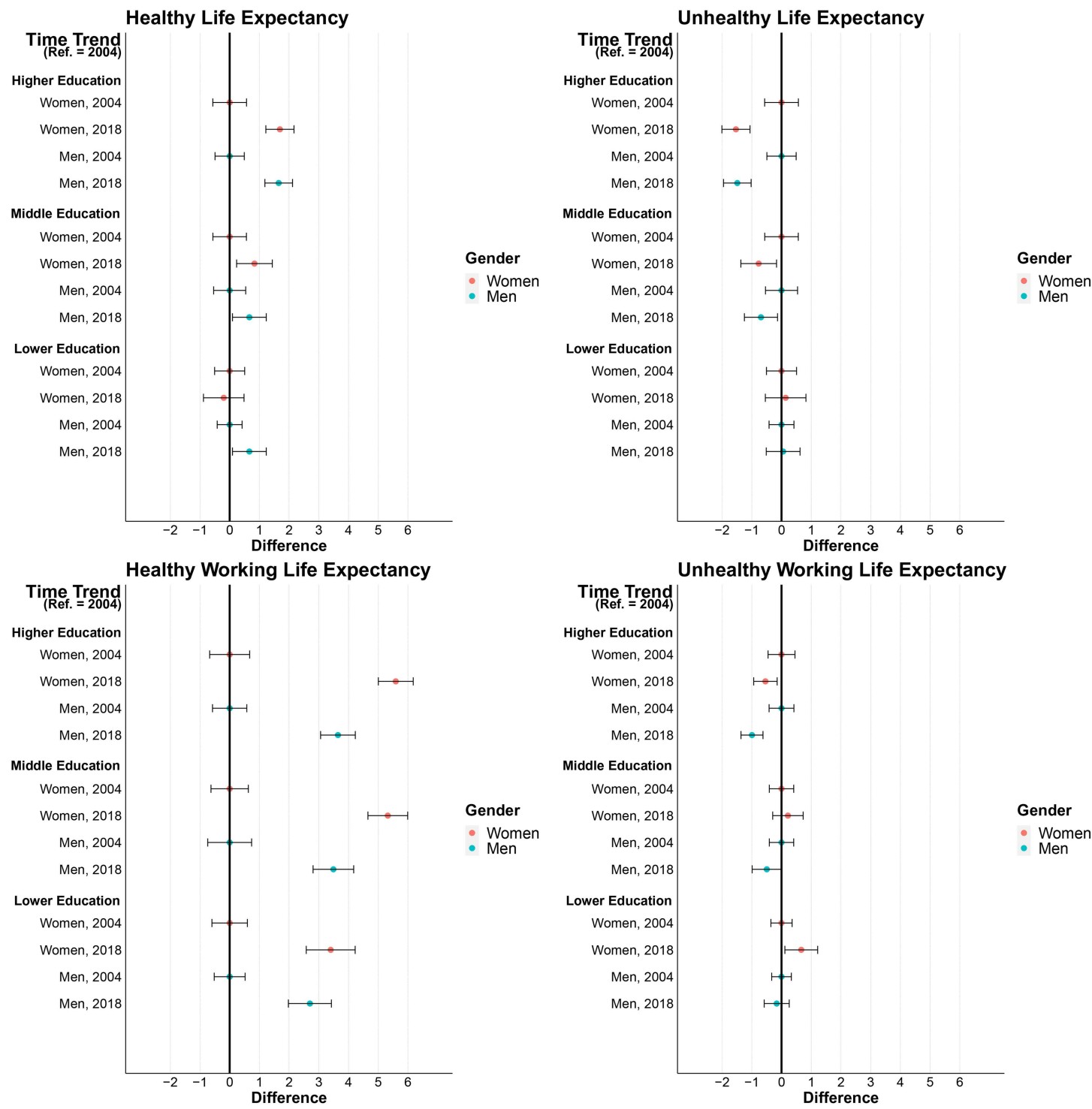

**Fig 4. Time trend in educational inequalities in partial Healthy Life Expectancy, Unhealthy Life Expectancy, Healthy Working Life Expectancy, and Unhealthy Working Life Expectancy between ages 30 and 69 in terms of Mental Health-related Quality of Life by year and gender.** Note: Health and Work Expectancies are given as partial life expectancies at age 30 up to age 69. 95% confidence intervals are given in brackets. 95% CIs were derived from the original (U)HLE and (U)HWLE but were shifted on the x-axis towards zero according to the difference value. Data Source GSOEP 2000–2020, authors'own calculations.

groups). UHWLE in terms of mHRQoL decreased significantly only in men and women with higher educational attainment, while it increased slightly in women with lower educational attainment (Fig 4).

## Discussion

### Main Findings

This is one of the few studies to analyse the evolution of educational inequalities in Healthy Life Expectancy in Germany over time, and the first one focussing specifically on educational inequalities in healthy years during working age. Furthermore, the study reports educational inequalities in Healthy Working Life Expectancy in Germany, taking into account inequalities in working-age mortality. We considered pHRQoL and mHRQoL as important indicators for occupational health. We found that educational inequalities in both healthy life years and healthy working life years developed more favourably among men and women with higher educational attainment. This holds for both pHRQOL and mHRQoL. At the same time, unhealthy life years and unhealthy working life years in terms of pHRQoL increased more strongly among individuals with lower educational attainment. With respect to mHRQoL, unhealthy (working) life years mostly decreased due to improving mHRQoL, but this decrease was stronger in men and women with higher education. These trends led to a clear widening in the gap in Health and Work Expectancies between the educational groups in Germany. The study also shows that the poorer trends in HLE in men and women with lower and middle educational attainment goes hand in hand with an increase in HWLE. This suggests that the gap between healthy years and healthy years spent working decreased over time and that the potential of further increase in working years is diminishing. This holds especially for people with lower educational attainment. They are falling behind and are less able to keep pace with the increase in retirement age from a health perspective.

### Findings in the context of previous research

We found rather constant (pHRQoL) or slightly increasing (mHRQoL) HLE and increasing HWLE in terms of both pHRQoL and mHRQoL in the total population. These findings are consistent with previous studies from Germany that found rather constant scores of pHRQoL and slightly increasing scores of mHRQoL during working age but increasing HWLE due to longer working lives [10,11]. From an international perspective, studies that analyse the development of educational inequalities in HLE are still relatively rare and do not focus explicitly on working age. Growing educational inequalities in HLE have been demonstrated for countries such as the US, Belgium, and the Netherlands using different health indicators and spanning different age ranges [40–42], which however supports the main findings of our paper on growing educational inequalities in H(W)LE. Our results differ from those of a previous study which found decreasing educational inequalities in HLE in terms of self-reported general health in women and rather constant inequalities in men between 1989 and 2009 in Germany [26]. Despite differences in health indicators, this could be interpreted as additional evidence of a deterioration in the health trends of people with a lower level of education. However, it should be kept in mind that this paper included all age groups from age 40 onwards [26] while our study focuses on the working-age population. This could be important, as it has been shown that trends in health over time differ between younger and older people in Germany [10,43].

Our paper shows that HLE in terms of pHRQoL and mHRQoL is higher than HWLE, but that the difference between them decreased over time due to the strong increase in working years. Similar findings were also reported by previous studies, which found decreasing differences between the number of healthy and working years [7] or a widening between HWLE and WLE [11] in the German population. Furthermore, it has been shown that working years spent in the presence of any chronic disease [17], of cardiovascular diseases [18] and of cancer [19] also increased over time, supporting our finding that the increase in HWLE is accompanied by an increase in UHWLE.

The literature is even more limited when it comes to trends in educational inequalities in HWLE. In line with our results, the few existing studies point to growing inequalities in the length of healthy working life [16,20,24]. Since there is no study

so far that investigated educational inequalities in HWLE based on pHRQoL or mHRQoL, our results on HWLE cannot directly be compared to previous studies focusing on the German population. Closest to our study, a previous paper investigated how educational inequalities in HWLE developed based on self-rated health (SRH) focusing specifically on older working age (50+) and ignoring the effects of mortality. The authors reported increasing inequalities in HWLE for older men and women over time, which is in line with our results [20].

A recent study found that there are considerable social inequalities in working-age mortality in Germany with cancer and cardiovascular diseases contributing most to inequalities in premature mortality [30]. The study supports our finding that not only health but also mortality aggravated the existing gap between SES groups in Health and Work Expectancies.

To counteract these inequalities, prevention should focus on both reducing health and mortality inequalities in the working-age population. An earlier study showed that low income, smoking, high body weight, and the parents' occupation are significant factors in explaining educational inequalities in Disability-free Life Expectancy [44]. Therefore, prevention strategies aiming at reducing these risk factors in the population may help to reverse the trend towards growing inequalities in HLE and HWLE. There are various approaches to improve the health of people with lower socioeconomic status, and many of them place greater responsibility on politicians or employers. They range from prevention in the workplace (e.g., occupational safety, health-conscious catering options, exercise in the workplace, training on work organisation and stress coping), and an upward adjustment of low wages, to structural approaches such as tax breaks for healthy foods, tougher smoking bans, or higher taxes on sugar, which are currently lacking in Germany. Multi-level approaches, i.e., the combination of structural approaches that reach all population groups more or less equally (such as tobacco bans or higher taxing) with interventions at the individual level (e.g., smoking cessation) has proven to be most effective, but should be "equity-checked" before implementation to prevent further exacerbating health inequalities [45,46]. The multifaceted nature of the approaches indicates that a health in all policies approach is needed in order to implement multi-level prevention measures that effectively reduce health inequalities [47].

## Strengths and limitations

Previous research suggests that LE in the GSOEP is overestimated due to the well-known health bias caused by health selection and an incomplete mortality follow-up [32]. We adjusted our estimates for the overestimation by adjusting the LE of the GSOEP to the level of the total German population. Note, however, that this does not affect the presented findings and conclusions. We decided for this adjustment to present realistic numbers of life years, which are in line with the official statistics on LE of the total population of Germany. It is possible that the underreporting of mortality differs between educational groups, which would lead to an overestimation of Health and Work Expectancies in the group with higher underreporting in the survey data compared to the total population. However, due to the lack of official data on educational inequality in life expectancy, we cannot determine whether the underreporting in mortality differs and, if so, to what extent, i.e., all assumptions in this regard are speculative. It should also be noted that the reported trends in educational inequalities remain unchanged, provided that the difference in underreporting between educational groups does not change over time, for which we have found no indication so far.

We applied the Sullivan method [34] as widely used approach to calculate Work and Health Expectancies. Unlike alternative approaches, this method is based on period life tables, which allowed us to adjust the LE of the GSOEP population to the overall German level.

Focusing on the length of healthy life in the general and working population, the study provides a comprehensive analysis of the development of working life from a health perspective. The analyses are based on the GSOEP, the largest and most comprehensive household sample in Germany, which combines detailed information on health and employment characteristics. This makes it possible to identify individuals with different combinations of health and employment status.

Another strength of the GSOEP is that it is one of the few data sources that contains information on mortality by SES for the German population. This information has already been used in earlier studies to determine LE in different SES

groups, e.g., [32,33]. These data characteristics allowed us to analyse Health and Work Expectancies taking into account educational inequalities in LE, which could not be considered in previous studies due to a lack of information on mortality by social status [20,27]. The advantage of our approach is that the age groups of older working age, which had to be excluded in earlier studies due to higher levels of mortality [27], could also be analysed. The inclusion of these age groups broadens the focus to older people who are particularly affected by the current (and future) increase in the retirement age. Our study reveals that LE between ages 30 and 69 varies by up to 1.6 years between educational groups, suggesting that inequalities in Health and Work Expectancies are underestimated when mortality is ignored.

We have chosen a cut-off value for the definition of unhealthy (working) life years that corresponds to a meaningful downward deviation in HRQoL from the national norm value. To ensure comparability of the analyses, this cut-off value is identical for all educational groups. However, it should be borne in mind that poor health in the low education group can have a more limiting effect on working capacity than in the higher education group due to different occupational requirements (e.g., for physical functioning). This further underscores the importance of public health efforts aimed at improving the health of this particularly vulnerable group of people. We conducted a sensitivity analysis to examine whether our results depend on the cut-off value used to define poor health. When applying a threshold value of 50, we obtained very similar results, showing that the finding of increasing educational inequalities in HLE and HWLE is robust to changes in the threshold value for the definition of poor health. This applies to both pHRQoL and mHRQoL (Table S3 and S4 in S1 File).

Last but not least, it must be noted that we investigated only two indicators of health. Health has many facets, and it is often the case that results based on one health indicator differ from those based on another one. In this paper, we have chosen two health indicators, which we see as being highly relevant for the ability to work. Nonetheless, future studies that investigate other health indicators would be very valuable to verify the presented results.

## Concluding Remarks

We found substantial educational inequalities in both healthy and healthy working life years in Germany in men and women. These inequalities widened over time due to more favourable developments in men and women with higher educational attainment. Our findings indicate that the health potential of future increases in working years is limited and that people with lower educational attainment are falling behind. Effective prevention strategies are needed to maintain the physical health of the working-age population. Special attention should be paid to individuals with lower educational attainment.

## Supporting information

**S1 File.** **Supporting Information including Table S1-S4 and Figure S1-S2.**
(PDF)

## Author contributions

**Conceptualization:** Juliane Tetzlaff, Fabian Tetzlaff, Marc Luy.

**Data curation:** Juliane Tetzlaff, Fabian Tetzlaff.

**Formal analysis:** Juliane Tetzlaff, Fabian Tetzlaff, Marc Luy.

**Methodology:** Juliane Tetzlaff, Fabian Tetzlaff, Marc Luy.

**Supervision:** Fabian Tetzlaff, Marc Luy.

**Validation:** Juliane Tetzlaff.

**Visualization:** Juliane Tetzlaff, Fabian Tetzlaff.

**Writing – original draft:** Juliane Tetzlaff.

**Writing – review & editing:** Fabian Tetzlaff, Marc Luy.

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
