## [Decision Letter · Decision Letter 0]

5 Oct 2025

Dear Dr. Tetzlaff,

We look forward to receiving your revised manuscript.

Kind regards,

Petri Böckerman

Academic Editor

PLOS ONE

Journal Requirements:

3. Please include captions for your Supporting Information files at the end of your manuscript, and update any in-text citations to match accordingly. Please see our Supporting Information guidelines for more information: http://journals.plos.org/plosone/s/supporting-information .

Additional Editor Comments:

The revised paper should address all comments.

Reviewers' comments:

Reviewer's Responses to Questions

**Comments to the Author**

1. Is the manuscript technically sound, and do the data support the conclusions?

Reviewer #1: Yes

Reviewer #2: Yes

2. Has the statistical analysis been performed appropriately and rigorously?

Reviewer #1: Yes

Reviewer #2: Yes

3. Have the authors made all data underlying the findings in their manuscript fully available?

Reviewer #1: No

Reviewer #2: Yes

4. Is the manuscript presented in an intelligible fashion and written in standard English?

Reviewer #1: Yes

Reviewer #2: Yes

Reviewer #1: The study relies solely on the SF-12 Physical Component Score (PCS), with a somewhat arbitrary cutoff of ≤40 for "poor health." This raises several issues:

No justification provided for the 40-point threshold beyond "clear downward deviation"

Physical health represents only one dimension of work capacity

Mental health, which significantly affects work ability, is excluded

Sensitivity analysis with alternative thresholds would strengthen conclusions.

Last comprehensive mortality follow-up was 2008, creating increasing uncertainty for later periods

Authors acknowledge higher underreporting in recent years but don't adequately address how this affects trend estimates

Differential mortality underreporting by education level could bias results

Discussion focuses heavily on German studies

Broader European comparison would enhance generalizability assessment

Reviewer #2: This manuscript addresses important issue of increasing disadvantage in health and work expectancies of individuals with lower educational level in Germany. This research work is original and has not been published elsewhere. The study also used a large data set to provide valuable insights into the trends over nearly two decades (2000-2020) and also applied a more commonly used approach, the Sullivan method. The study meets all applicable standards for the ethics and the article is written in Standard English. The findings are supported by strong statistics and are clearly broken down by gender and education level which aligns with prior literature. The paper also explains its own limitations, such as the possible bias in survey data and the fact that it only looks at one aspect of health.

As a recommendation for the authors to modify, although the study included a good health indicator, the exclusion of other dimensions of health and mental health aspects limits the full picture of work related inequalities. In the analysis, a more detailed sensitivity analysis regarding potential under reporting biases in mortality by SES might strengthen confidence in the estimates. Again, it would have also been great if the causal mechanisms behind increasing disparities such as occupational exposure, access to healthcare or lifestyle behaviors are addressed deeply in the discussion to guide the preventive mechanisms. Additionally, better if the recommendations for occupational health interventions and social policies to reduce inequalities explained in more detail rather than keeping it as” effective prevention strategies”.

**Do you want your identity to be public for this peer review?** For information about this choice, including consent withdrawal, please see our Privacy Policy

Reviewer #1: **Yes: ** SHOLA ADEYEMI

Reviewer #2: No

---

## [Author Response · Author response to Decision Letter 1]

15 Oct 2025

Dear Reviewer, dear Shola Adeyemi,

We thank you for your thoughtful comments, which we believe have contributed to improving the manuscript. We have made two important changes to the paper: First, we extended all analysis toward mental Health-related Quality of life to cover a broad picture of physical and mental HRQoL. Second, we performed sensitivity analysis for another cut-off value for good health. The results show that main finding of widening inequalities in healthy (working) life years holds also for mental HRQoL that the message of the paper remained the same.

Below, you will find our detailed response along with our changes and amendments made to the manuscript.

Reviewer 1

Comment 1: “The study relies solely on the SF-12 Physical Component Score (PCS), with a somewhat arbitrary cutoff of ≤40 for "poor health." This raises several issues: No justification provided for the 40-point threshold beyond "clear downward deviation". Physical health represents only one dimension of work capacity. Mental health, which significantly affects work ability, is excluded. Sensitivity analysis with alternative thresholds would strengthen conclusions”

Response 1: Thank you for this comment. We strongly agree that growing health inequalities in one health indicator do not necessarily have to be present in another health indicator. We already discussed this issue in the original draft. However, you have raised an important point here, namely that the analysis of physical health is best complemented by the analysis of mental health as both health domains can severely affect workability. We therefore decided to extend our analysis to the mental HRQoL. The results show that main finding of widening inequalities in healthy (working) life years based on physical HRQoL holds also for these additional analyses on mental HRQoL. We amended the results, methods, and discussion section accordingly. Please note, that we also checked all cited literature to be suitable for the amendments on mHRQoL. Based on your comment, we also added some sensitivity analysis using another cut-off value for good physical/mental physical HRQoL. These analyses show that the results are robust to changes in the threshold value. We added: “We conducted a sensitivity analysis to examine whether our results depend on the cut-off value used to define poor health. When applying a threshold value of 50, we obtained very similar results, showing that the finding of increasing educational inequalities in HLE and HWLE is robust to changes in the threshold value for the definition of poor health. This applies to both pHRQoL and mHRQoL (Table S2 and S3).” (lines 413-417)

We believe that setting the cut-off value to 40 (meaningful downward deviation from the population average value in Germany) is reasonable as this value is recommended in the literature and has already been used in other studies. However, we recognise that falling below this threshold may have a greater limiting effect on the ability to work for some educational groups than for others. Nevertheless, we consider the application of a uniform cut-off value for all educational groups to be important to maintain comparability (and as no recommendation for different cut-off values exist for the German population that are based on different thresholds of work capacity for lower and higher educated people). We added a few sentences to the Limitations section discussing this point: “We have chosen a cut-off value for the definition of unhealthy (working) life years that corresponds to a meaningful downward deviation in HRQoL from the national norm value. To ensure comparability of the analyses, this cut-off value is identical for all educational groups. However, it should be borne in mind that poor health in the low education group can have a more limiting effect on working capacity than in the higher education group due to different occupational requirements (e.g. for physical functioning). This further underscores the importance of public health efforts aimed at improving the health of this particularly vulnerable group of people.” (lines 407-413)

Comment 2: “Last comprehensive mortality follow-up was 2008, creating increasing uncertainty for later periods. Authors acknowledge higher underreporting in recent years but don't adequately address how this affects trend estimates. Differential mortality underreporting by education level could bias results.“

Response 2: Thank you for this thoughtful comment. We have given this point a further thorough consideration. We have already corrected the underreporting of mortality in the data as far as possible by applying a weighting procedure, which allowed us to provide more realistic values for (un-)healthy (working) life years. As the difference in partial life expectancy between the Survey population and the general German population (0.5 years in men and 0.2 years in women) is quite low (as is the general morality rate during working age), we assume that this underreporting have a rather minor impact on the results. Please noted that the reported trends in educational inequalities remain unchanged, provided that the difference in under-reporting between educational groups does not change over time. However, unfortunately, we currently have no indication as to whether the underreporting of mortality differs between educational groups and, if so, to what extent. Assumptions about the extent and direction of underreporting that would justify sensitivity analysis would therefore be highly speculative, which is why we have refrained from conducting sensitivity analyses. However, we agree that the possible consequences of differences in the strength of mortality underreporting between educational groups should be discussed in more detail. We therefore added “However, it is possible that the underreporting of mortality differs between educational groups, which would lead to an overestimation of Health and Work Expectancies in the group with higher underreporting in the survey data compared to the total population. However, due to the lack of official data on educational inequality in life expectancy, we cannot determine whether the underreporting in mortality differs and, if so, to what extent, i.e., all assumptions in this regard are speculative. It should also be noted that the reported trends in educational inequalities remain unchanged, provided that the difference in underreporting between educational groups does not change over time, for which we have found no indication so far.” (lines 380-387)

Comment 3 „Discussion focuses heavily on German studies. Broader European comparison would enhance generalizability assessment.”

Response 3: The paper focuses on the German population, which is why we have concentrated more on the German context so far. To enhance generalizability assessment, we extended the discussion towards international studies: “From an international perspective, studies that analyse the development of educational inequalities in HLE are still relatively rare and do not focus explicitly on working age. Growing educational inequalities in HLE have been demonstrated for countries such as the US, Belgium, and the Netherlands using different health indicators and spanning different age ranges [40-42], which however supports the main findings of our paper on growing educational inequalities in H(W)LE.”(lines 230-325) (…) “The literature is even more limited when it comes to trends in educational inequalities in HWLE. In line with our results, the few existing studies point to growing inequalities in the length of healthy working life [16, 20, 24].” (lines 343-345)

Reviewer 2

Overall Comment: “This manuscript addresses important issue of increasing disadvantage in health and work expectancies of individuals with lower educational level in Germany. This research work is original and has not been published elsewhere. The study also used a large data set to provide valuable insights into the trends over nearly two decades (2000-2020) and also applied a more commonly used approach, the Sullivan method. The study meets all applicable standards for the ethics and the article is written in Standard English. The findings are supported by strong statistics and are clearly broken down by gender and education level which aligns with prior literature. The paper also explains its own limitations, such as the possible bias in survey data and the fact that it only looks at one aspect of health.”

Overall Response: Thank you for your kind and constructive comments. Below you will find our responses and the changes we have made to the manuscript text.

Comment 1: “As a recommendation for the authors to modify, although the study included a good health indicator, the exclusion of other dimensions of health and mental health aspects limits the full picture of work related inequalities.”

Response 1: We appreciate your comment and expanded the analyses to include mental HRQoL in order to draw a more comprehensive picture of health in working age. Supporting the main message of our manuscript, the results show that main finding of widening inequalities in healthy (working) life years holds also for these additional analyses on mental HRQoL. We have amended the results, methods, and discussion section accordingly. Please note, that we also checked all cited literature to be suitable for the amendments on mHRQoL.

Comment 2: „In the analysis, a more detailed sensitivity analysis regarding potential under reporting biases in mortality by SES might strengthen confidence in the estimates.”

Response 2: We have given this point a further thorough consideration. We have corrected the underreporting of mortality in the data as far as possible by applying a weighting procedure, which allows us to provide more realistic values for (un-)healthy (working) life years. As the difference in partial life expectancy between the Survey population and the general German population (0.5 years in men and 0.2 years in women) as well as the general morality rate during working age is quite low, we assume that this could have a rather minor impact on the results. Please noted that the reported trends in educational inequalities remain unchanged, provided that the difference in under-reporting between educational groups does not change over time. Unfortunately, we currently have no indication as to whether the underreporting of mortality differs between educational groups and, if so, to what extent. Assumptions about the extent and direction of underreporting that would justify sensitivity analysis would therefore be highly speculative, which is why we have refrained from conducting sensitivity analyses. However, we agree that the possible consequences of differences in the strength of mortality underreporting between educational groups should be discussed in more detail. We therefore added “However, it is possible that the underreporting of mortality differs between educational groups, which would lead to an overestimation of Health and Work Expectancies in the group with higher underreporting in the survey data compared to the total population. However, due to the lack of official data on educational inequality in life expectancy, we cannot determine whether the underreporting in mortality differs and, if so, to what extent, i.e., all assumptions in this regard are speculative. It should also be noted that the reported trends in educational inequalities remain unchanged, provided that the difference in underreporting between educational groups does not change over time, for which we have found no indication so far.” (lines 380-387)

Comment 3: „Again, it would have also been great if the causal mechanisms behind increasing disparities such as occupational exposure, access to healthcare or lifestyle behaviors are addressed deeply in the discussion to guide the preventive mechanisms. Additionally, better if the recommendations for occupational health interventions and social policies to reduce inequalities explained in more detail rather than keeping it as ”effective prevention strategies””

Response 3 We agree that broadening the discussion in this direction is informative for readers. We therefore now refer to a study that identifies the risk factors promoting educational inequalities in HLE. Based on this study, we then discuss the various options for reducing health inequalities in HLE. The discussion highlights the importance of broader and equity-checked prevention approaches and the need for a health in all policies approach. We have expanded the discussion accordingly: “An earlier study showed that low income, smoking, high body weight, and the parents' occupation are significant factors in explaining educational inequalities in Disability-free Life Expectancy [44]. Therefore, prevention strategies aiming at reducing these risk factors in the population may help to reverse the trend towards growing inequalities in HLE and HWLE. There are various approaches to improve the health of people with lower socioeconomic status, and many of them place greater responsibility on politicians or employers. They range from prevention in the workplace (e.g., occupational safety, health-conscious catering options, exercise in the workplace, training on work organisation and stress coping), and an upward adjustment of low wages, to structural approaches such as tax breaks for healthy foods, tougher smoking bans, or higher taxes on sugar, which are currently lacking in Germany. Multi-level approaches, i.e. the combination of structural approaches that reach all population groups more or less equally (such as tobacco bans or higher taxing) with interventions at the individual level (e.g., smoking cessation) has proven to be most effective, but should be “equity-checked” before implementation to prevent further exacerbating health inequalities [45, 46]. The multifaceted nature of the approaches indicates that a health in all policies approach is needed in order to implement multi-level prevention measures that effectively reduce health inequalities [47].” (lines 357-372)

---

## [Decision Letter · Decision Letter 1]

5 Nov 2025

Lagging behind in health and work expectancies: Increasing disadvantage of individuals with lower educational level in Germany

PONE-D-25-38158R1

Dear Dr. Tetzlaff,

We’re pleased to inform you that your manuscript has been judged scientifically suitable for publication and will be formally accepted for publication once it meets all outstanding technical requirements.

Kind regards,

Petri Böckerman

Academic Editor

PLOS ONE

Additional Editor Comments (optional):

I am happy with the revised paper.

Reviewers' comments:

Reviewer's Responses to Questions

**Comments to the Author**

Reviewer #2: All comments have been addressed

2. Is the manuscript technically sound, and do the data support the conclusions?

Reviewer #2: Yes

3. Has the statistical analysis been performed appropriately and rigorously?

Reviewer #2: Yes

4. Have the authors made all data underlying the findings in their manuscript fully available?

Reviewer #2: No

5. Is the manuscript presented in an intelligible fashion and written in standard English?

Reviewer #2: Yes

Reviewer #2: I appreciate the thorough responses and improvements made in the manuscript. You have addressed my concerns.

Thank you.

**Do you want your identity to be public for this peer review?** For information about this choice, including consent withdrawal, please see our Privacy Policy

Reviewer #2: No

---

## [Editor Report · Acceptance letter]

PONE-D-25-38158R1

PLOS ONE

Dear Dr. Tetzlaff,

I'm pleased to inform you that your manuscript has been deemed suitable for publication in PLOS ONE. Congratulations! Your manuscript is now being handed over to our production team.

Kind regards,

on behalf of

Professor Petri Böckerman

Academic Editor

PLOS ONE